# Development of an International Database for a Rare Genetic Disorder: The *MECP2* Duplication Database (MDBase)

**DOI:** 10.3390/children9081111

**Published:** 2022-07-25

**Authors:** Daniel Ta, Jenny Downs, Gareth Baynam, Andrew Wilson, Peter Richmond, Aron Schmidt, Amelia Decker, Helen Leonard

**Affiliations:** 1Telethon Kids Institute, University of Western Australia, Perth, WA 6009, Australia; daniel.ta@telethonkids.org.au (D.T.); jenny.downs@telethonkids.org.au (J.D.); gareth.baynam@health.wa.gov.au (G.B.); andrew.wilson@health.wa.gov.au (A.W.); peter.richmond@uwa.edu.au (P.R.); 2Curtin School of Allied Health, Curtin University, Bentley, WA 6102, Australia; 3Rare Care Centre, Perth Children’s Hospital, Nedlands, WA 6009, Australia; 4Western Australian Register of Developmental Anomalies, King Edward Memorial Hospital, Subiaco, WA 6904, Australia; 5North Entrance, Perth Children’s Hospital, 15 Hospital Ave, Nedlands, WA 6009, Australia; 6Discipline of Paediatrics, School of Medicine, University of Western Australia, Perth, WA 6009, Australia; 7*MECP2* Duplication Foundation, Tuscon, AZ 85724, USA; aschmidt@mecp2d.org (A.S.); adecker@mecp2d.org (A.D.); 8Department of Paediatrics, University of Arizona College of Medicine, Tuscon, AZ 85724, USA

**Keywords:** *MECP2* duplication syndrome, rare neurodevelopmental disorder, intellectual disability, epilepsy, recurrent respiratory infections

## Abstract

The natural history of *MECP2* duplication syndrome (MDS), a rare X-linked neurodevelopmental disorder with an estimated birth prevalence of 1/150,000 live births, is poorly understood due to a lack of clinical data collected for research. Such information is critical to the understanding of disease progression, therapeutic endpoints and outcome measures for clinical trials, as well as the development of therapies and orphan products. This clinical information can be systematically collected from caregivers through data collation efforts—yet, no such database has existed for MDS before now. Here, in this methodological study, we document the development, launch and management of the international *MECP2* Duplication Database (MDBase). The MDBase consists of an extensive family questionnaire that collects information on general medical history, system-specific health problems, medication and hospitalisation records, developmental milestones and function, and quality of life (for individuals with MDS, and their caregivers). Launched in 2020, in its first two years of operation the MDBase has collected clinical data from 154 individuals from 26 countries—the largest sample size to date. The success of this methodology for the establishment and operation of the MDBase may provide insight and aid in the development of databases for other rare neurodevelopmental disorders.

## 1. Introduction

A rare disorder is defined as a condition that affects fewer than 1 in 2000 people as defined by the European Union in 2000 [1]. It is estimated that there are over 7000 rare diseases and about 250 new diseases are identified annually [2,3]. The diagnosis, management and development of orphan diagnostic and therapeutic products for rare disorders are often hampered by a lack of natural history information, validated biomarkers and outcome measures. One such disease is *MECP2* duplication syndrome (MDS; OMIM 300260), an X-linked rare neurodevelopmental disorder (RNDD) associated with intellectual disability and a complex profile of medical comorbidities which includes recurrent infections and seizures [4]. MDS is caused by a duplication of the methyl-CpG-binding protein 2 (*MECP2*) gene, of which loss-of-function mutations were first identified in Rett syndrome (RTT; OMIM 312750) in 1999 [5]. In the same year, the MDS syndromic phenotype was first described [6], but since then there have been relatively limited further natural history data. As such, MDS remains a poorly characterised RNDD. Our study using the Australian Paediatric Surveillance Unit (APSU) as one means of ascertainment to recruit 27 cases is, to our knowledge, the only epidemiological study of MDS to date, estimating a birth prevalence of 1 in 150,000 live births [7]. Prior to this current study, the largest study was a French case series featuring 59 males [8].

Data collection efforts which include patient databases and biobanks for RNDDs are important enablers of progress in the understanding of the natural history and genetic underpinnings of a disease before interventions that may impact the developmental trajectory of such disorders are trialled. Rare disease databases aid in establishing the epidemiology, diagnosis and life expectancy for a disorder, support patient recruitment for trials, and facilitate multi-centre collaboration [9]. RTT was described more than 50 years ago and has benefitted from a much longer research history than MDS [10], supported by the establishment of the population-based and longitudinal Australian Rett Syndrome Database (AussieRett) in 1993 amongst others [11]. Data have identified an incidence of ~1 in 10,000 female births [12], and survival of approximately two thirds of individuals to 38 years [13]. International rare disease databases are also critical to providing adequate case numbers even if these are not population representative. One example is the International Rett Syndrome Database (InterRett) which was established in 2002 and has been collecting questionnaire data from clinicians and families from over 50 countries, thus allowing for investigation of genotype–phenotype relationships [14]. Data from InterRett was seminal to the delineation of CDKL5 deficiency disorder (CDD; OMIM 300203) as a clinically distinct disorder since research by our group found less than one quarter of individuals with a pathogenic CDKL5 variant met the criteria for the early-onset seizure variant RTT [15]. These efforts led to our establishing the International CDKL5 Disorder Database (ICDD) in 2012 [16], and to further phenotyping of CDD including advances in understanding developmental milestones, functional skills and patterns of epilepsy [16,17,18].

Our comprehensive review of all case studies/series of individuals with MDS since 1999 identified some clinical areas that had received limited attention. We believed this could be addressed by the creation of an international registry for collection of more powerful and generalisable data for the disorder [4]. Cases of individuals with MDS have been ascertained alongside individuals with a *MECP2* mutation in the InterRett [19], thus forming an important platform for our development of the international *MECP2* Duplication Database (MDBase) in 2020. Following the success of InterRett and the ICDD [15,20,21], and informed by ongoing stakeholder engagement, our group created the MDBase in their likeness [9]. In this methodological paper, we illustrate the development of an international registry for MDS. Our aims were to (1) develop a family questionnaire to collect health-related data to address knowledge gaps; and (2) ascertain a large population of individuals with MDS for research. We reflect on the successes and limitations of our approach and make suggestions for the establishment of databases for other RNDDs.

## 2. Materials and Methods

### 2.1. Ethics Approval

Ethics approval was obtained from the University of Western Australia Human Research Ethics Committee (2019/RA/4/20/5929).

### 2.2. Formation of a Consumer Reference Group (CRG) and Identification of Key Research Areas

Parents of individuals with MDS previously ascertained through InterRett were contacted and invited to partake in a Consumer Reference Group (CRG) [20]. Additional members were sought by advertising on the *MECP2* Duplication Syndrome Family Talk Facebook group, the largest English-speaking online community for families of individuals with MDS. Thirteen mothers, seven living in the US and six in Australia, initially agreed to be a part of the CRG.

In April 2019, a teleconference was held with CRG members to discuss the gaps in knowledge including the lack of understanding of the natural history of the disorder as well as the need for developing outcome measures for clinical trials and qualitative studies of lived experiences of children and parents. We presented our initial goals to: (1) create an international MDS database and family questionnaire guided by the CRG for the recruitment of a large population of families; (2) further characterise the medical comorbidities of MDS to compare with and delineate from similar disorders such as Rett syndrome (RTT) and CDKL5 deficiency disorder (CDD) that hold phenotypic overlap; (3) examine the quality of life (QoL) of children with MDS and their caretakers and families; and (4) investigate how families cope with day-to-day challenges such as recurrent respiratory infections and epilepsy. With the approval and input of the CRG, we consolidated our research direction and included items in the questionnaire that reflected parents’ particular concerns relating to respiratory- and seizure-related illness as well as interventions for medical comorbidities affecting different systems of the body and quality of life of the individual and family.

### 2.3. Drafting a Family Questionnaire

The MDBase family questionnaire was based on the ICDD family questionnaire but modified for greater relevance to MDS [22]. Further domains were incorporated based on feedback from the CRG and a literature review in order to capture the full spectrum of the MDS phenotype. Various sections of the family questionnaire were piloted by members of the CRG as part of the process. The sections on Current Function, Cardiopulmonary, Epilepsy, Gastrointestinal Health, Ear, Nose and Throat (ENT), Medications, Hospital Admissions and Caregiver Quality of Life were piloted to ensure no components were missing and the questions were clear and worded in a sensitive manner. CRG members were emailed specific sections as a Microsoft Word document. Each section was piloted by at least three individuals between one to four times, with revisions based upon feedback from individuals. The final questionnaire is divided into two parts: Part 1 included modules about the child and Part 2 included modules about the caregiver and family (Table 1).

### 2.4. Database and Online Questionnaire Design

The MDBase infrastructure was created using FileMaker Pro, a database application, with access to the questionnaire for families provided through FileMaker WebDirect (Figure 1a). The entire family questionnaire was piloted by seven parents and seven researchers on a variety of devices including computers, tablets (iPad) and phones (Apple and Android), and web browsers (Safari, Chrome and Internet Explorer) to ensure optimal user experience. The questionnaire was designed so that families could enter their data, save their progress and return at a later time (Figure 1b). The database is hosted on a secure webserver located at the Telethon Kids Institute, Western Australia. Data are exported from the webserver to an internal password-protected FileMaker Pro database. Data are cleaned within the FileMaker Pro database, deidentified and exported to a Stata statistical software data file for analysis.

### 2.5. Advertising, Recruitment and Data Collection

Advertising material was created in the format of social media posts, video adverts and posters and distributed across various family support groups on Facebook (including the main *MECP2* Duplication Syndrome Family Talk group and sub-groups for females with MDS and families residing in the UK), Twitter and Instagram in the months leading up to the launch of the MDBase on 13 April 2020 and periodically every few months since. A dedicated webpage (https://rett.telethonkids.org.au/about/mecp2-duplication-syndrome/) on the Telethon Kids Institute website was created to distribute our advertising efforts. Advertisement flyers were distributed at the biannual *MECP2* Duplication Syndrome Family Conference (31 January–1 February 2020) with a link to register interest (https://rettregister.telethonkids.org.au/MECP2/Register). On the registration link, families were asked to select from which source they heard about the study (Figure 2).

Upon registration, families were contacted by phone or email (email only if no phone number was provided) and invited to participate in the MDBase and the family questionnaire. Previously ascertained families from InterRett were also contacted and invited to participate in the new database. Families of individuals with MDS who had died were also provided with the opportunity to contribute data. Families who agreed to partake in the study were emailed an information sheet and a set of secure, unique login credentials for each individual in the family of the person with MDS. Upon login, families were presented with a consent form before proceeding with their responses.

Families were called to ensure they received their log-in credentials and followed up via a courtesy call or email every one to three weeks for data collection until completion of the questionnaire or until the censor date (Figure 3). In our communication with families, we asked parents and caregivers to share our advertising material on social media and by word-of-mouth. In the months following the launch, we provided regular updates to families via social media (Facebook) on the number of registrations, response fraction and completion fraction of the family questionnaire and further encouraged participation from the community. All families in the MDBase also received regular updates on our research as well as new literature by other groups as part of an emailing list.

## 3. Results

From 13 April 2020 to 13 March 2022, attempts were made to contact the families of 254 individuals with an *MECP2* duplication known from prior sources (i.e., InterRett) and new sources (i.e., our registration webpage) via a combination of phone (*n* = 248) and email (*n* = 6; Figure 4). Initial successful contact was made with the families of 236 individuals of whom 205 (87%) agreed to participate in the questionnaire study. Aside from InterRett (*n* = 40), sources of ascertainment documented on the registration page were Facebook advertising (*n* = 119), the 2020 *MECP2* Duplication Syndrome Family Conference (*n* = 15), sharing of our study on MDS-related advocacy websites, by word-of-mouth (*n* = 18), and our study website (*n* = 13).

Questionnaire data was returned by the families of 154 individuals (response fraction of 75%) of which responses were fully completed by 138 (completion fraction of 67%). Of the 154 questionnaire responses, data providers were primarily biological mothers (123/136 [90%]), followed by biological fathers (7/136 [5%]), adoptive mothers (3/136 [2%]), grandparents (2/136 [1%]) and a sister (1/136 [1%]), and the relationship was unspecified for the remaining questionnaires. The median time taken to complete the questionnaire was 24 days (range: 0–273 days). Data were available for 134 males (87%) and 20 females (13%; Table 2), including two male sibling pairs and 15 individuals (13 male, 2 female) who had died prior to the census date. Genetic testing results were available for 128/154 (83%) individuals (109 males, 19 females), of which 49/128 (38%) individuals had an inherited *MECP2* duplication, 21/128 (16%) individuals had a de novo duplication and information was unavailable on mode of inheritance for 58/128 (45%) individuals. The largest proportion of individuals was from North America (*n* = 76 [49%]), followed by Europe (*n* = 51 [33%]), Oceania (*n* = 21 [14%]), Asia (*n* = 5 [3%]) and South America (*n* = 1 [1%]). The median age at data completion was 8.8 years for males (range = 0.9–51.6 years) and 10.2 years for females (range = 4.9–31.4 years).

Community engagement was instrumental in generating continued interest in and awareness of the project, with regular social media posts updating families with project milestones and news since the project was conceptualised in 2019 resulting in continual enrolment of families. On 20 March 2021, preliminary findings from the study on characterising medical comorbidities and their treatment were presented to families via a live online seminar hosted by Cure MDS, a non-profit fundraising organisation represented by families of individuals with MDS.

## 4. Discussion

The MDBase is the first international database for MDS, with a global reach that has currently enrolled the largest cohort of individuals with MDS in research at 154 individuals (134 males, 20 females). This case example successfully (1) identified gaps in clinical knowledge of MDS through a literature review with the consultation of stakeholders (families of individuals with MDS) to create a family questionnaire for the purposes of better characterising medical comorbidities and treatments, (2) developed, piloted and launched a user-friendly online questionnaire and database infrastructure and (3) ascertained a large population of individuals with MDS internationally for research.

The authors attribute the success of the MDBase to various factors. Firstly, stakeholder consultation and community engagement were critical to ensure research efforts were directed at issues most important to families and individuals with MDS. Stakeholder engagement can improve the relevance of research, increase stakeholder trust and enhance mutual learning by stakeholders [23]. We have endeavoured to involve stakeholders in the prioritization, implementation and dissemination stages of research. Additionally, endorsement by families through the sharing of our research via MDS-related advocacy websites and word of mouth accounted for approximately 10% of registrations, highlighting the strength of our consumer-driven approach. Secondly, the identification of a centralised online community of families representing individuals with MDS on social media (Facebook) for international sampling accounted for most of the registrations (approximately 60%). As such, for rare diseases such as MDS, we propose that social media such as Facebook is an important avenue for participant recruitment. Lastly, periodic but consistent advertising of the database on social media encouraged registration from new families. This strategy was supported by regular follow-up communication with families who agreed to participate in the study, to increase data capture.

The collection of health-related data through the MDBase family questionnaire provides a critical infrastructure for a more accurate understanding of comorbidities in MDS. These data have already identified a higher prevalence of pneumonia, bronchiolitis, bronchitis, gastroesophageal reflux and slow gut motility in males compared with females; as well as expanding the clinical phenotype of MDS by characterising ENT infections and UTIs and identifying a novel feature of urinary retention in both males and females [24].

Apart from the ongoing utilisation of this data to complete the research goals presented to our CRG, the value of this dataset lies in its potential use in the development and validation of outcome measures for MDS, which do not yet exist. Further data collection at later timepoints will provide longitudinal data and knowledge of the natural history for MDS but requires ongoing funding. Together, these points highlight the contribution of the MDBase to clinical trial readiness for MDS [25]. It is also important to highlight the benefit of remote data collection as a study model as the progress of this study has potentially been less impacted by the COVID-19 pandemic than other study designs that would require face-to-face data collection. This direction is increasingly recognised as an important strategy to substantially increase geographical and age range coverage in RNDD research [26].

The MDBase was designed as an international database to collect health-related information from families from across the world. Translation of the questionnaire into further languages is in progress but was prohibited by lack of resources when the database was first launched such that families only had access to an English version. Despite this, 35/154 (23%) of the respondents were from non-English-speaking countries. However, we acknowledge that there may have been some non-English-speaking families who did not have the means to complete the questionnaire resulting in a possible ascertainment bias. With additional resources, there are plans to translate the MDBase study questionnaire to other languages. Another limitation was that 13% of families who registered did not progress to questionnaire completion. This was because families may not have had the facilities (e.g., lack of access to a computer or a tablet) or time to complete the online questionnaire, or we could not establish further communication after initial contact and as such these families were not invited to partake in the study. Finally, the response fraction of 75% from the 205 families that initially agreed to partake in the family questionnaire study and completion fraction of 67% of 138/205 questionnaires could be due to the extensive length of the questionnaire as a result of its comprehensive nature.

Comparisons may be made between the MDBase and some of the few international registries other than InterRett and ICDD that collect clinical data on rare disorders associated with intellectual disability. Two such databases include the Global Prader–Willi Syndrome (PWS) Registry launched in 2015 and the Global Angelman Syndrome (AS) Registry launched in 2016 [27,28]. With an estimated prevalence of 1/10,000 to 1/30,000 for PWS [29], the Global PWS Registry recruited 1696 participants in the first four years of its operation with implementation of a research protocol of registry architecture and security, governance, questionnaire development, community engagement and data curation similar to ours [28]. Similarly, the research goal of the Global PWS Registry was to collect health-related data to better characterise and study the natural history of this disorder [28]. Research output stemming from the Global PWS Registry on issues such as weight problems, caregiver burden, suicidality, neuropsychiatric features, thrombosis risk and strabismus in PWS has been published, in the form of either retrospective studies from clinical and patient data collected via surveys (the largest of which featured data from 908 individuals) or prospective, observational studies recruiting individuals from the registry [30,31,32,33,34,35]. Such research output has been supported by funding from the Foundation for Prader–Willi Research [28]. Similar to the MDBase, the recruitment for the PWS Registry involves the use of social media, newsletters, collaboration with advocacy groups, advertising at family conferences and informational webinars [28]. Similar challenges faced by the PWS Registry utilising this research method include the limitations of parent/caregiver-reported data, incomplete survey/s, difficulties for families to answer questions for individuals with a complex medical history and engagement of older parents who may be less comfortable with web-based data collection [28]. The MDBase platform allows for clinical notes and records to be uploaded or emailed to researchers to supplement the parent/caregiver-reported information.

The authors of this paper recognise the importance of making optimal use of the data and returning results back to consumers and the community for their participation and for retaining active participation in the MDBase. As such, in addition to existing efforts detailed in this paper, we aim to provide regular research updates for families through webinars, newsletters and social media communication and disseminate publications stemming from the research. We also suggest that our research methodology is important for understanding the natural history of those individuals affected by rare diseases who are geographically isolated. Our methodological approach could be used to understand the natural history and undertake genotype/phenotype characterisation in other RNDDs such as Aristaless-related homeobox gene (ARX) spectrum disorders [36], Cornelia de Lange syndrome [37], Sotos syndrome [38], Pallister–Killian syndrome [39], and Rubinstein–Taybi syndrome [40], where to our knowledge no such databases have yet been established.

In conclusion, the establishment of MDBase, an international registry for MDS, is an important milestone in MDS research and marks the beginning of a comprehensive natural history study for this RNDD. The potential of the MDBase to contribute to clinical trial readiness is outlined in this methodological paper which highlights the success of the first two years of its launch, much of which is attributed to its consumer-driven approach, broad international-based recruitment and web-based data collection. The MDBase has currently collated the largest series of individuals with MDS to date and will continue to contribute to the clinical observation of the disorder in the hope that it will provide a clearer understanding of the natural history of MDS. Further data collection and cross-collaboration with hospital sites, clinicians, academic centres, scientists, patient advocacy groups and pharmaceutical and diagnostic industries could expand the value of our currently curated dataset, once an ongoing funding source has been established. An increased understanding of the clinical course and better help for patients affected by a rare disease will be facilitated by the establishment of databases and international cooperative networks.

## Figures and Tables

**Figure 1 children-09-01111-f001:**
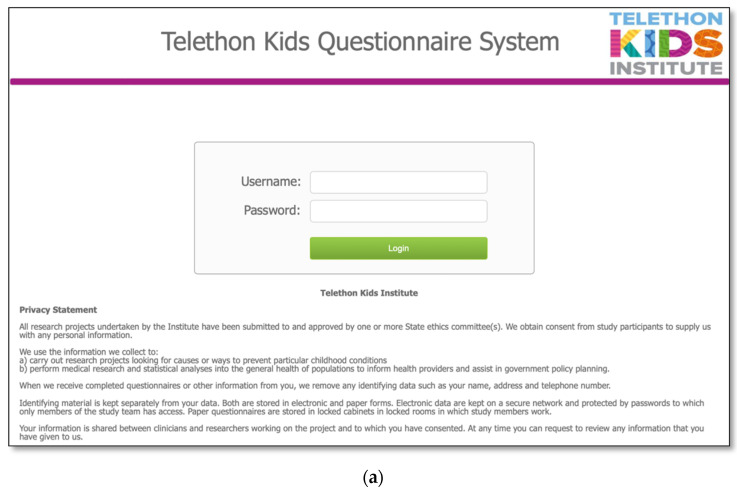
The International *MECP2* Duplication Database (MDBase): (**a**) login page and (**b**) progress status page.

**Figure 2 children-09-01111-f002:**
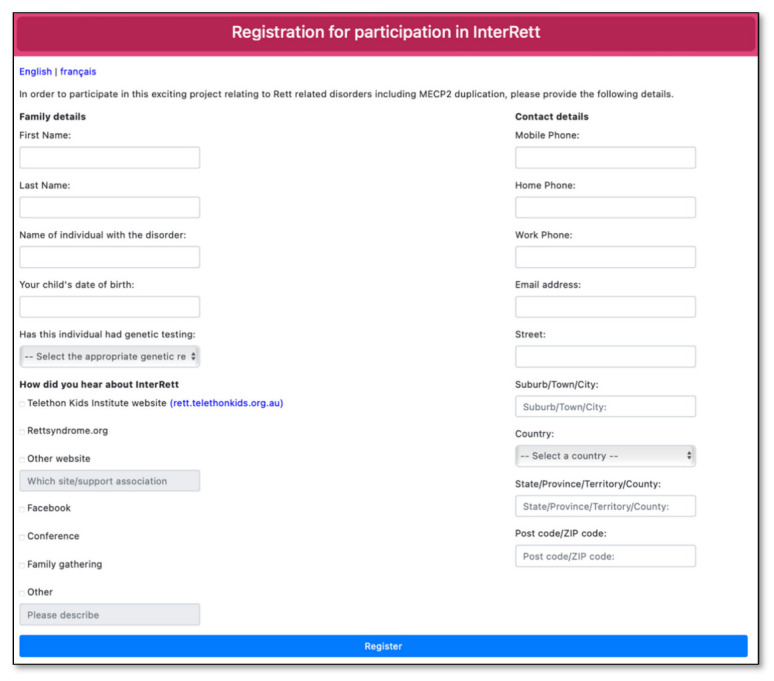
Registration page for the International *MECP2* Duplication Database (MDBase).

**Figure 3 children-09-01111-f003:**
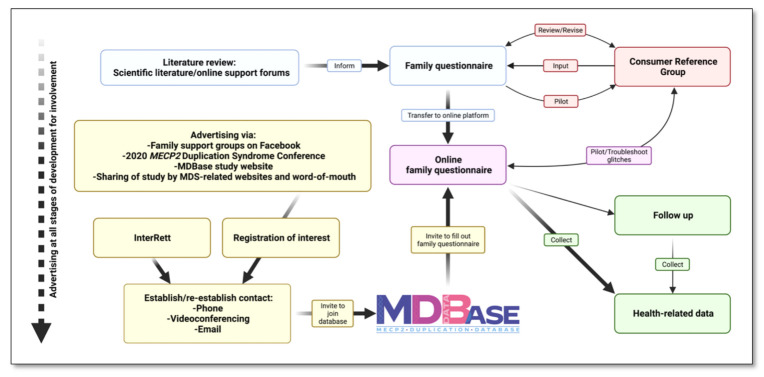
Development of the international *MECP2* Duplication Database (MDBase).

**Figure 4 children-09-01111-f004:**
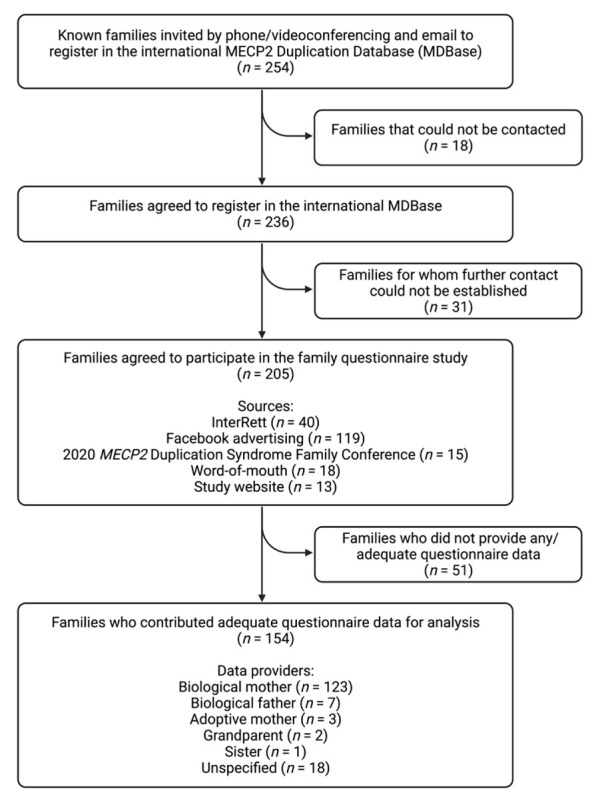
Study flow chart of case ascertainment and data collection for the international *MECP2* Duplication Database (MDBase).

**Table 1 children-09-01111-t001:** Overview of the international *MECP2* Duplication Database (MDBase) family questionnaire and associated instruments.

**Part 1: Questions About Individuals with *MECP2* Duplication Syndrome (MDS)**
Section 1: Early experiences
*Relating to pregnancy and period following birth of individual with MDS, first concerns, and diagnostic tests and examination surrounding MDS diagnosis*
Section 2: Early development
*Relating to gross motor, communication and social/emotional, and hand function developmental milestones*
Section 3: Regression
*Relating to the loss of any previously learned developmental skill*
Section 4: Current function
*Relating to current gross motor, communication, and hand function abilities*
Rett Syndrome Gross Motor Scale (RSGMS) Rett Syndrome Hand Function Scale (RSHFS)
Section 5: Cardiopulmonary
*Relating to conditions affecting the heart, lungs, and autonomic nervous system*
Section 6: Epilepsy
*Relating to seizures and/or epilepsy and their treatment*
Section 7: Gastrointestinal health
*Relating to gastrointestinal problems, feeding and surgical interventions*
Section 8: Ear, nose and throat (ENT)
*Relating to ear, nose and throat conditions and their treatment*
Section 9: Other medical conditions
*Relating to other medical conditions such scoliosis and bone fractures, urinary tract problems, and pain sensitivity*
Section 10: Sleep
*Relating to sleeping characteristics*
Sleep Disturbance Scale for Children (SDCS)
Section 11: Behaviours
*Relating to hand stereotypies, bruxism and other behaviours such as Rett syndrome-like and autism-like behaviours*
Rett Syndrome Behaviour Questionnaire (RSBQ) Developmental Behaviour Checklist Autism Screen Algorithm (DBC-ASA)
Section 12: Puberty
*Relating to sexual development*
Section 13: Medications
*Relating to medication history (prescribed, over the counter and natural medications as well as supplements)*
Section 14: Hospital admissions
*Relating to the length of time and reasons for hospital admissions*
Section 15: Current measurements
*Relating to current height, weight, and head circumference measurements*
Section 16: Day activities and school options
*Relating to schooling and issues surrounding schooling*
Section 17: Quality of life for individuals with *MECP2* duplication syndrome
*Relating to the quality of life of individuals with MDS*
Quality of Life Inventory—Disability (QI-Disability)
**Part 2: Questions About the Caregiver and Family**
Section 1: Family structure and demographics
*Relating to family circumstances for those caring for a child or adult with MDS*
Section 2: Family quality of life
*Relating to quality of life of immediate family*
Beach Centre Family Quality of Life (BCFQOL)
Section 3: Caregiver quality of life
*Relating to caregiver health and wellbeing*
Short Form Health Survey (SF-12) Depression, Anxiety and Stress Scale (DASS-21)

**Table 2 children-09-01111-t002:** Cohort characteristics of the international *MECP2* Duplication Database (MDBase) from data collected to March 2022.

Characteristics	n/N (%)
Gender	
Male	134/154 (87)
Female	20/154 (13)
Age at census date (years)	
0–<5	41/154 (27)
5–<10	42/154 (27)
10–<15	34/154 (22)
15–<20	14/154 (9)
20+	23/154 (15)
Mode of inheritance	
Inherited	49/128 (38)
De novo	21/128 (16)
Unconfirmed	58/128 (45)
Age at genetic diagnosis (years)	
<1	34/143 (24)
1–<2	34/143 (24)
2–<5	29/143 (20)
5–<15	31/143 (22)
15–<30	12/143 (8)
30+	2/143 (1)
Geographic distribution	
North America ^1^	76/154 (49)
Europe ^2^	51/154 (33)
Oceania ^3^	21/154 (14)
Asia ^4^	5/154 (3)
South America ^5^	1/154 (1)

^1^ USA and Canada. ^2^ UK, Italy, Netherlands, Spain, Poland, Germany, Norway, Denmark, Belgium, France, Finland, Sweden, Switzerland, Andorra, Hungary, Romania and Russia. ^3^ Australia and New Zealand. ^4^ Japan, Taiwan, India and United Arab Emirates. ^5^ Brazil.

## Data Availability

The data presented in this study are available on request from the corresponding author and subject to ethics approval.

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
