# Peer review of "Development of an International Database for a Rare Genetic Disorder: The MECP2 Duplication Database (MDBase)"

_children, 2022, doi:10.3390/children9081111_

Round 1
Reviewer 1 Report
In the submitted work Ta et al. present methodological study that documented the development, launch and management of the international MECP2 Duplication Database. This work is certainly valuable and may provide insight and aid in the development of databases for other rare neurodevelopmental disorders. The manuscript is nicely written. I recommend to accept the manuscript as it stands.
Author Response
Thank you very much for your feedback on our work - we greatly appreciate your comments and the time you have taken to read it.
Reviewer 2 Report
The research presented in the field of rare diseases is admirable. The authors have compiled a worldwide database with a great deal of effort over two years.
I have few methodological questions or comments:
Is it possible to estimate the number of potential participants? Exactly how many people were contacted and by what means? This is not really clear to me from the manuscript.
And -maybe I missed it in the text- the question if the survey was conducted only in English. Was the questionnaire only offered in English or was there, for example, also an offer in Spanish or Chinese. It is possible that this could lead to a certain bias in the sample.
And last but not least: Are there any considerations, for example, to conduct annual surveys among those affected?
But mentioned again: an interesting research paper. We do far too little work with groups with rare conditions.
Author Response
The research presented in the field of rare diseases is admirable. The authors have compiled a worldwide database with a great deal of effort over two years.
Thank you very much for your feedback – we appreciate the time you taken to provide this commentary.
I have few methodological questions or comments:
Is it possible to estimate the number of potential participants? Exactly how many people were contacted and by what means? This is not really clear to me from the manuscript.
It is difficult to estimate the number of potential participants for a range of reasons. We have previously estimated that the birth prevalence of MECP2 Duplication syndrome is 1/150,000 live births.1 By comparison with an estimated prevalence of 1/15,000 to 1/30,0002 3 the Global Prader Willi Syndrome Registry recruited 1,696 participants in the first four years of its operation. Similarly with a birth prevalence of 1/9,000 females for Rett syndrome4 the International Rett syndrome Database (InterRett) holds data on approximately 1800 genetically confirmed cases. However, given the much lower birth prevalence of MECP2 Duplication syndrome, we anticipate the number of participants in MDBase will be relatively fewer.
Additionally, we have provided revisions to the text on page 7 to explain how many families were contacted (n = 254) and added Figure 4 to explain the step-wise process of ascertaining families from contact to data collection.
And -maybe I missed it in the text- the question if the survey was conducted only in English. Was the questionnaire only offered in English or was there, for example, also an offer in Spanish or Chinese. It is possible that this could lead to a certain bias in the sample.
Unfortunately, the questionnaire was only offered in English as we did not have capacity for multiple translations. Although nearly a quarter of respondents completing the questionnaire were from non-English backgrounds, we acknowledge the risk of ascertainment bias occurring in the future if translations are not made available particularly for under-resourced families. We have provided revisions starting on line 272 to address your very valid point.
And last but not least: Are there any considerations, for example, to conduct annual surveys among those affected?
We agree the about the importance of conducting follow-up surveys to provide longitudinal . We address this point starting on line 260, indicating that for us, the provision of resources will allow for this collection of data at future data points.
But mentioned again: an interesting research paper. We do far too little work with groups with rare conditions.
- Giudice-Nairn P, Downs J, Wong K, et al. The incidence, prevalence and clinical features of MECP2 duplication syndrome in Australian children. Journal of Paediatrics and Child Health 2019;55(11):1315-22 doi: 10.1111/jpc.14399[published Online First: Epub Date]|.
- Lionti T, Reid SM, White SM, et al. A population-based profile of 160 Australians with Prader-Willi syndrome: trends in diagnosis, birth prevalence and birth characteristics. Am J Med Genet A 2015;167A(2):371-8 doi: 10.1002/ajmg.a.36845[published Online First: Epub Date]|.
- Whittington JE, Holland AJ, Webb T, et al. Population prevalence and estimated birth incidence and mortality rate for people with Prader-Willi syndrome in one UK Health Region. J Med Genet 2001;38(11):792-8 doi: 10.1136/jmg.38.11.792[published Online First: Epub Date]|.
- Fehr S, Bebbington A, Nassar N, et al. Trends in the Diagnosis of Rett Syndrome in Australia. Pediatric Research 2011;70(3):313-19 doi: 10.1203/PDR.0b013e3182242461[published Online First: Epub Date]|.